# Screening of Membrane Protein Production by Comparison of Transient Expression in Insect and Mammalian Cells

**DOI:** 10.3390/biom13050817

**Published:** 2023-05-11

**Authors:** Jagan Mohan Kaipa, Ganna Krasnoselska, Raymond J. Owens, Joop van den Heuvel

**Affiliations:** 1Curexsys GmbH, Annastrasse 27, 37075 Goettingen, Germany; 2Department of Biomedical Sciences, University of Copenhagen, Blegdamsvej 3B, 18.5, 42, 2200 Copenhagen, Denmark; 3Structural Biology Division, Wellcome Centre for Human Genetics, University of Oxford, Roosevelt Drive, Oxford OX3 7BN, UK; 4Rosalind Franklin Institute, Harwell Campus, Didcot OX11 0QX, UK; 5Helmholtz Center for Infection Research, Department of Structure and Function of Proteins, Inhoffenstrasse 7, 38124 Braunschweig, Germany

**Keywords:** membrane proteins, transient gene expression, insect cells, human embryonal kidney (HEK) cells, Chinese hamster ovary (CHO) cells, fluorescence-detection size-exclusion chromatography

## Abstract

Membrane proteins are difficult biomolecules to express and purify. In this paper, we compare the small-scale production of six selected eukaryotic integral membrane proteins in insect and mammalian cell expression systems using different techniques for gene delivery. The target proteins were C terminally fused to the green fluorescent marker protein GFP to enable sensitive monitoring. We show that the choice of expression systems makes a considerable difference to the yield and quality of the six selected membrane proteins. Virus-free transient gene expression (TGE) in insect High Five cells combined with solubilization in dodecylmaltoside plus cholesteryl hemisuccinate generated the most homogeneous samples for all six targets. Further, the affinity purification of the solubilized proteins using the Twin-Strep^®^ tag improved protein quality in terms of yield and homogeneity compared to His-tag purification. TGE in High Five insect cells offers a fast and economically attractive alternative to the established methods that require either baculovirus construction and the infection of the insect cells or relatively expensive transient gene expression in mammalian cells for the production of integral membrane proteins.

## 1. Introduction

Membrane proteins play a vital role in many physiological and pathological processes. Understanding the structure and functions of these proteins is essential for the development of novel diagnostic and anti-infective agents by utilizing a structure-guided drug design approach. The purification of eukaryotic membrane proteins is challenging due to the usually low expression levels and instability of membrane proteins after extraction and purification in detergent-containing solutions. The choice of expression system and method of gene delivery strongly influences the production and quality of authentically folded membrane proteins.

Both mammalian and insect cells have been adapted for use in membrane protein production using different gene delivery methods [1,2,3,4,5,6]. Due to the availability of a functional cellular machinery for translation, proper folding, and post-translational modifications, mammalian membrane proteins, including receptors, ion channels, and transporters, are widely produced in mammalian cells, such as Chinese hamster ovary CHO and human embryonal kidney HEK293 cell lines. Moreover, mammalian cells provide a near-native lipid milieu that can be vital for the biogenesis of active membrane proteins [7]. Therefore, stable, and transient mammalian expression systems are extensively used as effective methods to produce membrane proteins for functional and structural analysis. However, the incomplete or heterogeneous glycosylation of receptors has been reported following over-expression in mammalian cells [8]. The use of a glycosylation-deficient derivative of Expi293, Expi293 *Gnti*- cell line was reported to improve the stability of the produced proteins [9]. However, numerous studies addressing receptor folding, trafficking, and function require native N-linked glycosylation patterns of proteins [10,11]. Furthermore, efficient gene delivery into mammalian cells can be achieved using lentivirus-mediated transduction or baculoviral infection using BacMam [12]. However, this requires a labor-intensive multi-step process. Although mammalian cells have been successful in expressing a wide range of membrane proteins [13], their use in drug discovery, target screening, and structural analysis has been limited by relatively high production costs and low production yields.

The baculovirus expression vector system (BEVS) in insect cells provides a powerful eukaryotic production system for membrane proteins. Baculovirus-infected insect cell expression systems use *Sf*9/*Sf*21 cells, derived from the ovarian tissues of *Spodoptera frugiperda* (fall armyworm) or BTI-TN-5B1-4 (High Five) cells from *Trichoplusia ni* (cabbage looper) [14,15,16]. The very strong late viral promoters that drive the expression from baculoviruses enable relatively high levels of secreted as well as intracellular protein to be obtained. However, in the case of integral membrane proteins, the production may be limited by the disassembly of the secretion machinery and the loss of cellular structures during the infection process [17]. As an alternative and to avoid the time-consuming generation of baculoviruses, virus-free transient gene expression in *S. frugiperda* insect cells using plasmid-based vectors has been developed [18,19,20,21]. This initially resulted in only low yields of the expressed protein in *Sf* cells using the fusion promoter of hr5 and IE1 of the autographa californica multiple nucleopolyhedra virus (acMNPV), until the TGE method was largely improved by Bleckmann et al. [22,23,24], Shen et al. [25], Farell et al. [26], Mori et al. [27], and Puente-Massaguer et al. [28,29] using High-Five cells in combination with the IE2 promoter of orgyia pseudotsugata MNPV. Here, a fast and simple virus-free expression system was developed by introducing the strongest available RNA polymerase II-dependent promoter for the insect cells pOpIE2 and improving the experimental parameters to a robust and efficient method to produce intra-cellular and secreted proteins [24].

To date, very few side-by-side, detailed comparisons of protein expression systems have been reported for specific classes of proteins, such as transporters [30] or G protein-coupled receptors (GPCRs) [31]. Typically, expression systems are chosen case by case based on prior knowledge of homogeneity, post-translational modifications, functionality, pharmacology, and cytotoxicity [32,33,34,35,36].

In the current work, we compare viral and non-viral gene expression systems for the production of full-length eukaryotic membrane proteins in either mammalian or insect cell lines. We chose six eukaryotic membrane proteins for this study that included two GPCRs and membrane proteins with different subcellular localization (Table 1). Firstly, we compare the well-established baculovirus expression vector system (BEVS) in *Sf*9 and ExpiSf9 for the fast screening of the membrane protein expression with transient gene expression (TGE) in High Five cells, mammalian HEK293 cells and CHO cells and the lentiviral transduction of HEK293 cells. The ExpiSf9, Expi293F, and ExpiCHO cell lines were chosen as they are easy-to-acquire expression cell lines from a commercial source available for academic research.

In the second part of the project, we focus on the performance of the virus-free transient gene expression (TGE) method in High Five cells in comparison with transient expression in HEK293 mammalian cells and the BEVS in *Sf* cells for the production of membrane proteins. Protein production was monitored by the fusion of the target proteins to either GFP or mCherry, enabling the detection of protein expression in the cells and the analysis of the quality of purified protein using fluorescence-detection size-exclusion chromatography (FSEC). The results of the small-scale protein production trials show that the transient protein expression in both Expi293F mammalian cells, grown at the reduced temperature of 30 °C, as well as the High Five insect cells followed by the solubilization of the membrane fraction in the commonly used mild non-ionic n-dodecyl-ß-D-maltopyranoside/cholesteryl hemisuccinate (DDM/CHS) micelles yielded homogeneous membrane proteins. The additional analysis of the purification strategy showed that affinity capture on a Strep-Tactin^®^ matrix by fusion to a Twin-Strep-tag substantially increased the sample quality of the purified membrane proteins compared to immobilized metal ion affinity chromatography (IMAC) using a hexahistidine tag.

## 2. Materials and Methods

### 2.1. Cloning the Genes of Interest

The genes of interest (GOI) were cloned in the multi-target pOPINE vector developed at the Dr. R. Owens lab [37] (Addgene, Watertown, MA 02472, USA, plasmid #26043) using the ClonExpress II One Step Cloning kit (Vazyme Distributionspartner, Kassel, Germany, #C112) and the In-Fusion^®^ cloning technique with GOI fusion via an HRV3C protease cleavage site to a C-terminal eGFP-His6 fusion and placed under the control of the very-late baculoviral p10 promoter (*Sf*9 insect cells) and CAG promoter (mammalian cells). For the lentivirus-mediated gene expression in HEK293S GnTI–TetR cells, the above-mentioned GOI-HRV3C-eGFP-6His constructs were subcloned into the transfer pHR-CMV-TetO2 plasmid [5] (kindly provided by Dr. Benjamin Bishop) under the control of the CMV promoter. For the transient gene expression in High Five insect cells, the genes were inserted into the vector pOpiE2-C7-GOI-mCherry-3C-Twin-Strep^®^ (this paper) under the control of the immediate early OpIE2 promoter and fused at the C-terminus to mCherry-3C-Twin-Strep^®^ and also into the vector pOpIE2-C9 with the C-terminal HRV3C-eGFP-6His tag.

### 2.2. Cell Culture

*Spodoptera frugiperda Sf*9 (Thermo Fisher Scientific, Bremen, Germany, #11496015) and ExpiSf (Thermo Fisher Scientific, Bremen, Germany, #A35243) cells were maintained in autoclaved 250 mL plastic VITLAB wide-mouth bottles (Thermo Fisher Scientific, Bremen, Germany, #10407363) sealed with sterile BRAND gas-permeable foil (Merck, Darmstadt, Germany, #BR701365) in antibiotic-free Gibco^TM^ Sf-900^TM^ II and ExpiSf CD media (Thermo Fisher Scientific, Bremen, Germany, #10902104 and #A3767803, respectively) at 27 °C on an orbital shaker rotating at 180 rpm (Insect incubator 44R, Eppendorf New Brunswick, Merck, Darmstadt, Germany). Cells were kept at a density of 0.5–5.5 × 10^6^ cells/mL and passaged every 3–4 days in a fresh medium.

*Trichoplusia ni* cells (High Five, BTI-TN-5B1-4) obtained from Invitrogen were cultivated in the EX-CELL^®^ 405 medium (Sigma-Aldrich #24405C, Merck, Darmstadt, Germany) at 27 °C in 250 mL sterile vented Corning^®^ Erlenmeyer cell culture flasks on an orbital shaker rotating at 120 rpm (Orbitron, Infors HT, Hamburg, Germany). Cells were passaged every 2 or 3 days by diluting to 0.5 × 10^6^ cells/mL to maintain exponential growth.

Expi293F cells (Thermo Fisher Scientific, Bremen, Germany, #A14527) were maintained in the Gibco Expi293^TM^ expression medium (Thermo Fisher Scientific, Bremen, Germany, #A1435102) in 125 mL vented Nalgene™ single-use PETG Erlenmeyer flask with a plain bottom (Thermo Fisher Scientific, Bremen, Germany, #4115-0125) at 37 °C, 5% CO_2_, and 120 rpm in a humidified orbital shaker (2066XLL, N-Biotek, Republic of Korea). The passaging of the cells was performed every 4 days and the cells were maintained between 0.5 and 5.5 × 10^6^ cells/mL.

### 2.3. Construction of Baculoviruses and Expression in Sf9 and ExpiSf Cells

One day before transfection, *Sf*9 cells (passage number 4–20) were seeded at 0.5 × 10^6^ cells/mL in a 24-well deep-well block (Qiagen, Hilden, Germany, #19583) (1.5 mL each well) and allowed to grow overnight to reach ~1 × 10^6^ cells/mL. No antibiotics were added to the cells at this step. Transfection-grade plasmids (0.5–1.0 mg) were purified from 150 mL overnight in an LB culture using the Qiagen Plasmid Plus Midi Kit (Qiagen, Hilden, Germany, #12945) and, if required, stored at −20 °C in sterile 1.5 mL Eppendorf tubes. The recombinant baculoviruses were obtained using a single-step flashback approach. In the reaction mixture, each pOPINE-GOI-3C-eGFP-His6 baculovirus transfer plasmid (500/750 ng) was added to *Bsu*36I (New England Biolabs, Frankfurt-am-Main, Germany, #R0524S), linearized empty DsRed bacmid (500 ng) (the bacmid was a kind gift from Prof. Ian Jones) diluted in 100 µL Sf-900 II SFM (Gibco^TM^ Sf-900^TM^ II, Thermo Fisher Scientific, Bremen, Germany) following the addition of 5 µL FuGENE HD Transfection Agent (Promega, Walldorf, Germany, #E2312). The transfection cocktail was mixed thoroughly, for 30 min at RT, and then gently added to the *Sf*9 cells. The cultures were incubated at 27 °C and 180 rpm for 7 days. The culture medium (P0 stock) was harvested and used for another round of re-infection (P1-production). For P1 amplification, 10 µL of P0 stock (1:300 dilutions) was used to infect 3 mL of *Sf*9 cells at 1 × 10^6^ cells/mL. After incubation for 7 days, the plates were centrifuged at 1000× *g* for 10 min at RT to spin down the cells and cell debris. The supernatant (P1 stock) was removed and used for the last round of amplification. P2 virus stock was generated by infecting 50 mL of *Sf*9 cells with 0.5 mL P1 stock. The resulting supernatant was collected through centrifugation at 1000× *g* at RT, sterile-filtered, titered, and stored at +4 °C for 3–4 months.

Titres of P2 viral stocks were determined using the end-point dilution approach. Briefly, *Sf*9 cells (3 mL) were seeded at a density of 2 × 10^6^ cells/mL in a 24-well deep-well block and infected with 100 µL virus stocks prepared as 1:6000, 1:3000, 1:600, 1:300, 1:100, and 1:60 dilutions. After 3 days of incubation (27 °C, 220 rpm), 20 µL aliquots of cells were analyzed in Tali™ Image Cytometer. The number of cells expressing DsRed was used as an indication of infection. The titers obtained from the endpoint dilution assay were used to determine the quantity of virus (multiplicity of infection, MOI) to be added to the cultures for expression tests in *Sf*9 and ExpiSf cells. On the day before infection, the cells were seeded to a density 1 × 10^6^ (*Sf*9) and 2 × 10^6^ (ExpiSf) cells/mL respectively and then the following day infected at MOI of 5 or 10. Cultures were incubated for 3 days at either 27 °C or 32 °C with shaking at 180 rpm.

### 2.4. Transient Gene Expression in Insect High Five Cells

The transient transfection in High Five cells was conducted as described earlier [28]. In brief, logarithmically grown High Five cells were harvested by centrifugation at 180× *g* for 4 min at RT and were resuspended in fresh EX-CELL^®^ 405 (Sigma-Aldrich, Merck, Darmstadt, Germany) media to a cell density of 4 × 10^6^ cells/mL. The linear 40 kDa polyethylenimine (PEI 40 kDa) transfection stock solution was prepared by dissolving 1 mg PEI 40 kDa (Polysciences Europe GmbH, Hirschberg an der Bergstrasse, Germany, #24765) into 1 mL sterile, deionized nuclease-free water. The PEI stock solution was adjusted to pH 7.0, sterilized by filtration, and stored at −20 °C. The resuspended cells were transfected with recombinant plasmids (1 µg/10^6^ cells) and linear PEI 40 kDa (Polysciences, Europe GmbH, Hirschberg an der Bergstrasse, Germany, #24765) at a ratio of 1:4. The expression vector was directly added to the cells (1 µg DNA per 1 × 10^6^ cells). Without delay, the PEI 40 kDa transfection stock solution was also added directly to the cells (4 µg PEI 40 kDa per 1 × 10^6^ cells). The cells were placed on an orbital shaker for 3 h at 27 °C, 120 rpm (Orbitron, Infors-HT, Hamburg, Germany). Three hours post-transfection, the cells were diluted 4-fold to a final density of 1 × 10^6^ cells/mL with pre-warmed EX-CELL^®^ 405. The transfected cell culture was further incubated and monitored daily by light microscopy to estimate the cell growth and analyzed by flow cytometry (Guava easyCyte mini, MerckMillipore, Darmstadt, Germany) for eGFP and/or mCherry fluorescence. For maintaining exponential growth, the culture volume was doubled 48 h post-transfection by the addition of fresh EX-CELL^®^ 405 media (preincubated at RT). The cells were harvested after 72 h by centrifugation (3000× *g* for 20 min at RT).

### 2.5. Transient Gene Expression in Mammalian Expi293F Cells

Transient gene expression in Expi293F cells was performed as described earlier [38]. As an alternative, the protocol described by the manufacturer of the Expi293^TM^ expression system can be used, which may result in higher transfection efficiencies. However, for financial reasons, we used the affordable PEI40 established protocol for transfection. Briefly, Expi293F cells were centrifuged at 500× *g* for 10 min and resuspended in pre-warmed Gibco Expi293™ expression medium (Thermo Fisher Scientific, Bremen, Germany, #A1435102) to reach a final density of 2.5 × 10^6^ cells/mL. Separately, the transfection mixture was prepared by mixing 2.5 µg DNA diluted in 250 µL Gibco OPTI-MEM serum-free medium (Thermo Fischer Scientific, Bremen, Germany, #31985070) with 20 µg of PEI 40 kDa (Polysciences, Europe GmbH, Hirschberg an der Bergstrasse, Germany #24765) for each mL of the Expi293F culture (2.5 × 10^6^ cells/mL). After thorough mixing, the transfection mixture was incubated for 10 min at RT. The transfection cocktail was gently added to the Expi293F cells and placed in a shaking mammalian cell incubator (2066XLL, N-Biotek, Republic of Korea) at either 37 °C or 30 °C, 125 rpm, 5% CO_2_ with 80% humidity. Twenty h post-transfection, the culture media were supplemented with a final concentration of 5 mM valproic acid, 6.5 mM sodium propionate, and 0.9% glucose to boost the protein expression. Valproic acid and sodium propionate are known histone deacetylase inhibitors [39,40] and are used to cope with transcriptional repression of transfected plasmids. The use of valproic acid, sodium propionate, and glucose feed in combination helps to enhance gene expression substantially. The transfected cell culture was further incubated and monitored daily by light microscopy to estimate the cell growth and analyzed by flow cytometry (Tali™ Image Cytometer) for eGFP fluorescence. The cells were harvested after 6 days by centrifugation (3000× *g* for 10 min).

### 2.6. Transient Expression in ExpiCHO Cells

The propagation, expression, and transfection of ExpiCHO cells were performed using the ExpiCHO™ Expression System kit (Gibco™, Thermo Fisher Scientific, Bremen, Germany, #A29133). One day before transfection, the cells were seeded in a 6-well tissue culture plate (Greiner Bio-One, Kremsmünster, Austria, #657160) with 3 mL/well each, to reach the density of ~6 × 10^6^ cells/mL on the following day. For each transfection reaction, in parallel, 6 µg DNA was diluted in 120 µL ice-cold OptiPRO™ SFM Complexation Medium and 10 µL ExpiFectamin™ CHO reagent was diluted in 110 µL ice-cold OptiPRO™ SFM Complexation Medium. Two parts were mixed and incubated 5 min at RT. The mixture was added dropwise on top of the cells and the plate was incubated for 1 day at 37 °C, 150 rpm, and 5% CO_2_ in a humidified shaking incubator (mammalian cell incubator 2066XLL, N-Biotek, Republic of Korea). On the next day, the cells were supplemented with 18 µL ExpiFectamine™ CHO Enhancer and 480 µL ExpiCHO™ Feed (protocol for Max Titer Protocol). The temperature was shifted to 30 °C (according to the Max Titer Protocol) or kept standard (37 °C) as described by the manufacturer. On day 5, another 480 µL ExpiCHO™ Feed was added. The expression was stopped on day 7 (37 °C) or day 12 (30 °C).

### 2.7. Construction of Lentiviruses and Expression in HEK293S (GnTI-TetR) Cells

The lentivirus production and transduction of HEK293S GnTI–TetR cells followed the established protocol by Elegheert et al. [5]. In brief, one day before transfection, Lenti-X 293T cells (Takara Bio Europe SAS, Saint-Germain-en-Laye, France, #632180) were diluted 1/6 in a DMEM/F12 medium containing 10% FBS and seeded overnight in a 6-well tissue culture plate (Greiner Bio-One, Kremsmünster, Austria, #657160), with 2 mL of cells in each well. On the day of transfection, the cells were approximately 50–60% confluent. The old medium was decanted, the cells were washed with PBS, and 2 mL of DMEM/F12 containing 2% FBS was added. For each transfection mixture, 1.3 µg of the transfer plasmid pHR-CMV-TetO2 containing a sub-cloned construct, 1.3 µg of envelope plasmid (pMD2.G; Addgene, Watertown, MA 02472, USA, #12259), and 1.3 µg of packaging plasmid (psPAX2; Addgene, Watertown, MA 02472, USA, #12260) were pre-mixed in 35 µL of serum-free DMEM in a sterile 1.5 mL Eppendorf tube. In parallel, 10 μL of polyethyleneimine 25 kDa branched (PEI 25, Merck, Darmstadt, Germany, #408727) stock was mixed with 35 µL of DMEM in a sterile 1.5 mL Eppendorf tube. The two solutions were added together, thoroughly mixed, and incubated for 10 min at RT following dropwise addition to separate the wells of the tissue culture plate. The plate was incubated for 3 days at 37 °C and 5% CO_2_. The supernatant containing lentiviral particles (~2 mL) was collected with a sterile syringe and filtered through a 40 µm sterile cell strainer (Sigma-Aldrich, Merck, Darmstadt, Germany, Corning #352340) to remove the cell debris. The lentiviral stock was supplemented with 2 mL fresh DMEM/F12/10% FBS and 2.5 µL polybrene (10 mg/mL of stock solution, Merck TR-1003-G) and was used to transduce HEK293S GnTI–TetR cells. Expression in the lentivirus-infected HEK293S GnTI–TetR cells was induced by the addition of 10 μg/mL of doxycycline hydrochloride (10 mg/mL of stock, Merck D3447) and was held for 3 days at 37 °C or 6 days at 30 °C.

### 2.8. Efficiency of Gene Delivery and in-Cell GFP Fluorescence

To quantify the number of viable cells expressing GFP and the magnitude of GFP fluorescence signals, the suspension cell-based assay using Tali™ Image-Based Cytometer (Thermo Fischer Scientific, Bremen, Germany, #T10796) was used according to the Invitrogen application notes Tali™ Image-Based Cytometer: “Quantitating GFP & RFP Expressing Cells with the Tali Image Cytometer’’ and “Measuring Viability in GFP Expressing Cells with the Tali Image Cytometer”. For each measurement, only fresh samples of cells (20 µL) were imaged on the slides supported by the Tali™ Image-Based Cytometer using ≥9 fields for statistical accuracy. As recommended by the manufacturer, the fluorescent cells were separated from the autofluorescent cells by setting a minimum fluorescence value (threshold) on the histograms generated from the cell data by the Tali™ cytometer. To correct data for cells and media background autofluorescence, we also used the negative control (cells transfected with a construct lacking a reporter gene). To prevent the debris in the sample from being analyzed, the cell size gate on the Tali™ Image-Based Cytometer was used, allowing the instrument to include only the cells of interest in the downstream fluorescence analysis. The average cell size for each cell line was estimated by the Tali™ cytometer: Expi293F, 9.7 ± 0.5 µm; ExpiCHO, 14.2 ± 0.4 µm; ExpiSf (virus-infected), 17.0 ± 0.6 µm; Sf9 (virus-infected), 17.0 ± 1.7 µm; and High Five, 21.0 ± 1.5 µm. We used the suggested cell size gating (>5 µM) to eliminate data from the cell debris. This size gate fits the literature data on the average size of the cell debris in the range of 2–5 µm [41]. For each condition, 6 transfection reactions were measured, and the data are presented as mean ± S.E.

### 2.9. Protein Purification

Cell pellets (from 100 mL expression culture) were resuspended in 15 mL of ice-cold buffer containing 50 mM of Tris pH 8.0, 150 mM of NaCl, and 5 mM of MgSO_4_ with a freshly added inhibitor cocktail for mammals (Sigma-Aldrich, Merck, Darmstadt, Germany P8340) and DNase I (Sigma-Aldrich, Merck, Darmstadt, Germany, SLBW0018). The resuspended cells were disrupted on ice by sonication for 12 min in 50% duty cycle with 10% amplitude and sonication pulse duration of 10 s with 10 s pause (MSE Soniprep 150 Plus Ultrasonic Disintegrator). The unbroken cells were removed by centrifugation (Megafuge 40R, Thermo Fisher Scientific, Bremen, Germany) at 4000× *g* for 35 min. The supernatant was subjected to ultracentrifugation at 150,000× *g* for 3 h (Optima L-80 XP Ultracentrifuge, Beckmann Coulter, Krefeld, Germany) to spin down the membrane fraction. The membrane pellet obtained after ultracentrifugation was mechanically re-suspended with Dounce homogenizer in buffer 50 mM of Tris pH 8.0, 150 mM of NaCl, and 5 mM of MgSO_4_ supplemented with 10% glycerol. All steps were performed at 4 °C unless otherwise stated. The homogenized crude membranes were either directly used for reconstitution and purification or stored at −80 °C.

The membrane fusion proteins were solubilized from the crude membrane fraction using 1% (*v*/*v*) DDM (n-dodecyl-β-D-maltopyranoside) [36,38] for 1 h on a vertically rotating platform in a cold room with a subsequent 1 h ultracentrifugation at 100,000× *g* (Optima L-80 XP Ultracentrifuge, Beckmann Coulter, Krefeld, Germany). The resulting supernatant was applied to either pre-equilibrated resins Ni-NTA^®^-Sepharose 6 Fast Flow (Cytiva, Freiburg, Germany) for His6 constructs and Strep-Tactin^®^ superflow (IBA-Lifesciences, Göttingen, Germany) for Twin-Strep^®^ constructs in the batch mode. After gently mixing overnight at 4 °C using a vertical rotating platform, the resins were allowed to settle by gravity sedimentation and the supernatant discarded. The resins were washed 3 times with 10 CV of buffer containing 50 mM of Tris pH 8.0, 150 mM of NaCl, 5 mM of MgSO_4_, and 1% (*v*/*v*) DDM at pH 8.0. In the case of Ni-NTA^®^ purification, 35 mM of imidazole was also added to the wash buffer. To elute the bound proteins, 300 µL of buffer containing either 400 mM of imidazole for His6-protein purification or 10 mM of desthiobiotin for Strep-Tactin^®^ purification. Resins were incubated for 1 h with gentle agitation and finally the elution fractions were collected by gravity.

### 2.10. Fluorescence-Detection Size-Exclusion Chromatography

A total fo 110 µL of purified samples were transferred to Chromacol 0.3 mL Screw Top Fixed Insert Vial (Thermo Fisher Scientific, Bremen, Germany, 03-FISV) with Thermo Fisher Scientific 9 mm Autosampler Vial Screw Thread Caps. The injection of samples (10–100 µL) on a SRT-C SEC-300 HPLC column (20 mL, Sepax Technologies, Inc., Newark, Delaware, USA) was conducted automatically using the high-throughput auto-sampler of the HPLC system (Shimadzu UK Ltd., Milton Keynes, UK) with a Shimadzu RF20A Fluorescence Detector. The samples were run at 0.5 mL/min flow rate.

### 2.11. Gel Electrophoresis and In-Gel Fluorescence

A total of 10 µL of protein sample was mixed with 10 µL of 2× SDS PAGE gel-loading buffer (100 mM of Tris, pH 6.8, 4% *w*/*v* SDS, 0.2% *w*/*v* of bromophenol blue, 10% *v*/*v* of β-mercaptoethanol, and 20% *v*/*v* of glycerol). Then, the sample was incubated at 37 °C for 30 min. After incubation, the sample was loaded onto precast SDS-PAGE gel (any kD Mini-PROTEAN, Bio-Rad) and run at 120 V (constant voltage) for 90 min at 4 °C until the dye reached the bottom of the gel. The gel was carefully removed from the cassette and placed in deionized water. The visualization of protein bands in the gel was performed either by a 50 mL solution of Coomassie blue stain (InstantBlue, Abcam ISB1L) incubated for 60 min on an orbital shaking platform (Polymax 2040, Heidolph Instruments GmbH Schwabach, Germany) with subsequent washing in milliQ water or directly visualized in a fluorescent gel imager (Typhoon FLA 9000, GE Health care) in the GFP channel (489 nm exc/508 nm em)

## 3. Results

### 3.1. Gene Delivery

The expression of the six selected membrane proteins (Table 1) fused to green fluorescent protein (GFP) was analyzed under different transfection conditions in three mammalian (Expi293F, HEK293S GnTI-TetRm and ExpiCHO) and three insect cell lines (*Sf*9, ExpiSf, and High Five).

For transient expression, the cell cultures were infected with either recombinant baculoviruses (BEVS in *Sf*9 and ExpiSf cells) or transfected with recombinant plasmids using PEI 40K (TGE in High Five and Expi293F cells) and ExpiFectamine CHO (TGE in ExpiCHO cells) (Table 2). The cells were grown in suspension in chemically defined media, with the exception of HEK293S GnTI-TetR cells, which were grown adherent in the serum-containing media.

The efficiency of the gene delivery methods was estimated by counting the total and fluorescent cells using a Tali™ Image-Based Cytometer. From these measurements, the fractions of the viable cells expressing GFP were determined (see Figure 1). Furthermore, the effect of the cultivation temperature (Table 2) on the transfection efficiency was monitored.

For baculovirus-infected cells, DsRed integrated into the baculoviral genome was used to monitor the transfection efficiency of the viruses. The highest number of cells expressing GFP was obtained for the insect cells (*Sf*9 and ExpiSf cell lines) infected with high-titre baculoviruses (at MOI = 10, Appendix A). On average, more than 70% of *Sf*9 cells expressed GFP after viral infection. No difference between the fraction of the infected cells grown at either 27 °C or 32 °C was observed. The good correlation between DsRed and GFP transfection ratio (>80% of cells show both signals) indicated that most of the infected cells expressed the GFP fusion protein.

The plasmid-based TGE in Expi293F cells typically produces a transfection efficiency of 60–80%; however, for the membrane proteins, lower but reproducible transfection efficiencies were observed with a maximum at day 3 of 20–25% at 30 °C and ~30% at 37 °C across multiple experiments. The ExpiCHO cells were transfected using ExpiFectamine CHO, following the manufacturer’s recommended protocol, and this resulted in less than 10% of cells with a detectable GFP expression. The plasmid-based TGE in the High Five insect cells showed higher transfection efficiencies compared to mammalian TGEs at day 3 and were in the range of 30–45%.

The expression of the GFP-fused targets in different cells was assessed by fluorescence microscopy (see Figure 2 and Appendix A). The results show that GFP signals were detected for all six targets, but the levels varied significantly between different cell hosts. The low cultivation temperature of 30 °C improved viability and increased the membrane protein expression in Expi293F (see Appendix A) considerably. A relatively poor expression was observed for the Golgi-apparatus-membrane-localized hUGTrel8 and bNTCP in all cell hosts, except Expi293F and High Five. The expression of all targets in ExpiCHO cells (days 3–6) resulted in generally modest levels of expression and, therefore, these cells were not used for further analysis in this study.

To assess whether the GFP signals correspond to the expression of intact membrane proteins, detergent extracts were generated and analyzed on SDS-PAGE by in-gel fluorescence. Fluorescent bands corresponding in size to full-length proteins (65–96 kDa range for full-length membrane protein–eGFP fusion proteins) (see Table 3) were observed for all the targets but with variable intensities in the extracts from the different expression hosts. The presence of an additional band corresponding to free GFP appeared to be both host- and protein-dependent. However, full-length proteins were obtained for all targets expressed by TGE in Expi293F at 30 °C (Figure 3, lane 1) and High Five cells at 27 °C (Figure 3, lane 7). Furthermore, a substantial breakdown was observed following the doxycycline-induced expression of rNTSR1, hUGTrel7, and hUGTrel8 targets in lentivirus-transduced HEK293S GnTI-TetR and, additionally, no fluorescence for hA2AR was visible for this expression system (Figure 3, lane 3). hA2AR has been reported to be susceptible to the proteolytic cleavage of the C-terminus [42]. The two bands observed for hGAT1 may represent different states of protein glycosylation (Figure 3, lane 6).

Given the relatively poor quality of the targets expressed by lentivirus-transduced HEK293S GnTI-TetR cells compared to the other cell expression systems, this method was not pursued for the analysis of the expressed proteins (see also Appendix A).

### 3.2. Fluorescence-Detection SEC Analysis of Membrane Fusion Proteins Extracted and Purified in DDM

To compare the homogeneity of the membrane target proteins obtained after purification from different expression hosts, the GOI-3C-eGFP-His6 constructs were produced in 100 mL cultures of the mammalian and insect expression systems. After 3–6 days of expression, the cell pellets were collected, disrupted by sonication, and a total membrane fraction solubilized in 1% (*v*/*v*) of n-dodecyl-β-D-maltopyranoside (DDM). The proteins were partially purified by binding and elution from Ni-NTA resin. The estimated amount of the purified protein after the initial capture by affinity purification is presented in Table 4. Additionally, the samples were analysed by fluorescence-detection size-exclusion chromatography (FSEC).

Firstly, we compared the FSEC profiles of the samples extracted and partially purified in DDM, following expression in either Expi293F and *Sf*9 cells. The Expi293F cells were grown at either 30 °C or 37 °C following transfection; the ExpiSf cells were grown at either 27 °C or 32 °C after infection with baculoviruses. Three of the targets (rNTSR1, bNTCP, and hGAT) produced monodisperse single peaks following solubilisation and partial purification in DDM from membranes of both Expi293F cells and ExpiSf cells grown at 30 °C and 32 °C, respectively, whereas hA2AR produced two discrete peaks that may correspond to a mix of monomers and dimers (Figure 4 and Figure 5). By contrast, the samples of these proteins expressed in Expi293F grown at 37 °C or ExpiSf cells grown at 27 °C produced heterogeneous FSEC traces, indicating that, for these proteins, culture temperature rather than expression host appears to be the key factor determining protein homogeneity. With the exception of hUGTrel7 purified from Expi293F grown at 30 °C, the FSEC traces for hGUTrel7 and hUGTrel8 produced in both mammalian and insect cells showed that the purified samples were polydisperse, indicating either a misfolding or aggregation of the proteins. The FSEC profiles of the purified membrane fusion protein samples from BEVS in *Sf*9 cells showed largely comparable results. A higher growth temperature of 32 °C favoured the production of better-quality samples (sharper elution peak with fewer shoulders with an increase in monodisperse material for rNTSR1, dNTCP, and hGAT1) (Appendix A).

### 3.3. Transient Gene Expression of Membrane Fusion Proteins in High Five Insect Cells and DDM/CHS Extraction

The plasma membrane of the insect cells has a low cholesterol content compared to the mammalian cells, and the addition of cholesteryl hemisuccinate (CHS) to the DDM detergent extraction of proteins expressed in insect cells has been shown to increase sample homogeneity [36]. Therefore, the six membrane proteins expressed in insect cells either introduced by baculovirus infection or transient transfection were solubilized and purified in DDM/CHS. The FSEC analysis showed that there was no improvement in the quality of the proteins extracted from baculovirus-infected cells (Figure 5 and Figure 6). However, the FSEC profiles of the samples produced by transient gene expression in High Five insect cells closely matched those from transient Expi293F cells at 30 °C in DDM, even showing a higher proportion of a monodisperse hUGTrel7 species for TGE in High Five (Figure 4 and Figure 6). Collectively, the results show that, by screening expression in both insect and mammalian cells, homogeneous samples of five out of the six target membrane proteins fused to GFP were obtained, providing a basis for further protein production at a larger scale. Thus, transient gene expression in insect cells (High Five) offers a cost-effective alternative to the transient gene expression of membrane proteins in mammalian cells.

### 3.4. Comparison of Twin-Strep and Hexahistidine Tags for the Affinity Purification of Membrane Proteins

The purity of the six membrane targets recovered from DDM/CHS-solubilized membranes from TGE in High Five cells by IMAC was compared to Twin-Step-Strep-Tactin affinity purification. To distinguish the Twin-Strep and His6 affinity purification, genes for the six target proteins were inserted into a vector, adding mCherry as a fluorescent protein and the Twin-Strep tag to the C-terminus in a configuration comparable to the eGFP-His6 fusion constructs described previously. The six membrane fusion proteins were produced in a 100 mL culture volume of High Five cells and harvested 3 days post-transfection. The purified proteins were analyzed using SDS-PAGE to assess the integrity of the bands on the stained gels to confirm their purity. The yield and purity of the samples recovered using Twin-Strep purification with Strep-Tactin beads was greater than that using Ni-NTA affinity chromatography (Figure 7B,C and Table 4; Appendix A). The FSEC profiles of the Twin-strep-tagged mCherry rNTSR1, bNTCP, hGAT1, and hA2A proteins were similar to the GFP-His-tagged proteins (Figure 7). However, the hUGTrel7 and hUGTrel8 samples were more heterogenous, indicating that the higher expression led to aggregation and mis-folding. In general, these targets proved more difficult to express homogeneously than the other membrane proteins in the test set.

## 4. Discussion

We compared some aspects of eukaryotic membrane protein production in insect and mammalian cells with respect to transfection efficiencies, in-cell and in-gel protein signals, the resulting protein quality, and the total protein yield. By means of this comparison, we assessed the applicability of the recently established protocol for the transient virus-free expression of eukaryotic membrane proteins in the suspension cultures of High Five cells. Although the transfection efficiency of High Five cells did not exceed baculovirus-mediated insect cell transfection, generally high levels of transfection up to 45% using linear cationic polymer PEI 40K in High Five cells were obtained. This is, on average, 20% higher than the transfection levels achieved for the cultures of Expi293F cells with PEI40 transfection. However, the transfection of Expi293F cells may be improved by using Expifectamine 293 according to the protocol developed by the manufacturer for the transfection of Expi 293F cells.

The FSEC analysis of the partially purified proteins showed that the choice of cell line and expression temperature variously affected protein integrity (effect of T), protein yield (effect of T and cell line), and protein monodispersity in the solution (effect of T and cell line), depending on the target protein. n-Dodecyl-β-D-maltopyranoside (DDM) was used in both the solubilization and purification of the test set of membrane proteins, as it is generally the first-choice detergent in the production of membrane proteins for structural studies [36,42,43,44,45], often combined with CHS cholesteryl hemisuccinate (CHS) [46]. However, despite its mild and non-denaturing properties, DDM or DDM/CHS is not universal for protein extraction and purification, and some proteins display heterogeneity and tend to aggregate in DDM micelles and require other detergents (lauryl maltose neopentyl glycol (LMNG), Digitonin, and others) to improve the stability/solubility of the samples [47]. In contrast to the plasma membranes of the mammalian cells, those of insect cells contain tenfold less cholesterol and high levels of phosphatidylethanolamine and phosphatidylinositol, but no detectable phosphatidylserine and glyco- and sphingolipids [43,48,49], which may also be essential for protein function [50].

The native oligomeric state of target membrane proteins used in our study require further investigations. Experimental data indicate that both monomeric and dimeric forms can be physiologically active. Many GPCRs are known to form obligate heterodimers (e.g., GABA receptor GABBR1/GABBR2) or multimers that may be involved in additional regulatory mechanisms [51,52]. Furthermore, there is substantial evidence for the homo- and hetero-oligomerization of members of the class A GPCR subfamily [53,54,55,56]. However, in the case of rNTSR1 and hA2AR, homo-dimerization in native membranes and detergent solutions appears less evident [57,58]. Human and rat NTSR1 have previously been produced in baculovirus-infected insect cells for structural determination. The proteins purified in either LMNG/CHS or Digitonin micelles with the addition of ligands behaved as monomers [59,60,61,62,63,64]. In our experiments, rNTSR1 was purified as a monomeric protein from the membranes of all tested insect cell lines, either baculovirus-infected *Sf*9 and ExpiSf cells or transiently expressed in High Five cells. Samples of rNTSR1 purified in DDM and DDM/CHS without the addition of ligands were highly homogeneous in solution. rNTSR1 was also successfully produced in native membranes of Expi293F cells at 30 °C. Human hA2AR has also been produced in baculovirus-infected insect cells for structural analysis and behaved as a monomer in either octylthioglucoside (OTG), OTG/CHS, or DDM/CHS [65,66]. Another study showed that samples of hA2AR produced in *S. cerevisiae* were purified as a mixture of oligomers, dimers, and monomers. The homodimers of hA2AR could be separated from other oligomeric forms and remained stable in a solution for days [67]. In the current study, hA2AR appeared monomeric by FSEC, although a small fraction of larger oligomers was also observed in the FSEC profiles.

*SLC10* family members are glycoproteins expressed in the plasma membrane and form homodimers for proper regulation and expression [67]. Currently, the human *SLC10* family comprises seven members, of which NTCP (*SLC10A1*) and apical sodium-dependent bile acid transporter (ASBT) (*SLC10A2*) are the best characterized members, and both are Na+/taurocholate cotransporters. Structural information has only been obtained for the bacterial homologs of ASBT [68,69], whereas the structure of human NTCP produced in insect cells has recently been reported [70]. Previously NTCP had been shown to dimerize, but the functional unit remained the individual monomer of NTCP [71]. In this study, a small portion of what appeared to be dimers was also present in samples obtained from baculoviral-infected *Sf*9 and ExpiSf cells at 27 °C. However, samples extracted in DDM and/or DDM/CHS from all other expression conditions were exclusively monomeric.

The human genome encodes many members of the solute carrier 6 (*SLC6*) gene family, including GABA-transporters GAT1 encoded by *SLC6A1* and the *SLC35* family. Human UGTrel7 encoded by *SLC35D1* is located in the endoplasmic reticulum (ER) in mammalian cells and was the first nucleotide-sugar transporter (NST) identified to transport two kinds of nucleotide sugars [72]. Members of *SLC35* family are likely to form multiprotein complexes with other transporters [73]; however, the functional oligomeric state of hUGTrel7 and hUGTrel8 is yet to be confirmed. In the case of hUGTrel8 encoded by *SLC35D2,* the crystal structure of the yeast Vrg4 homologue is available [74]. The data show that Vrg4 is monomeric in decylmaltoside (DM) and shows signs of dimerization upon reconstitution in liposomes. Neither hUGTrel7 or hUGTrel8 produced completely monodisperse products following expression in either mammalian or insect cells. The transient expression of hUGTrel7 in High Five cells at 27 °C and purified by IMAC in DDM/CHS provide the most encouraging results for this class of membrane protein.

## 5. Conclusions

The use of small-scale expression screening of recombinant membrane proteins to identify conditions for subsequent scale-up has become well established. For membrane proteins, fusion to a fluorescent reporter protein provides a highly sensitive way of monitoring expression and assessing protein quality without the need for extensive purification. Using this approach, a systematic comparison of production of six representative membrane proteins in insect and mammalian cells was conducted. The results demonstrate that TGE in High Five insect cells offers an alternative to the established methods that require a multi-step process for baculovirus construction and the infection of insect cells with comparable results to transient gene expression in mammalian cells for the production of integral membrane proteins.

## Figures and Tables

**Figure 1 biomolecules-13-00817-f001:**
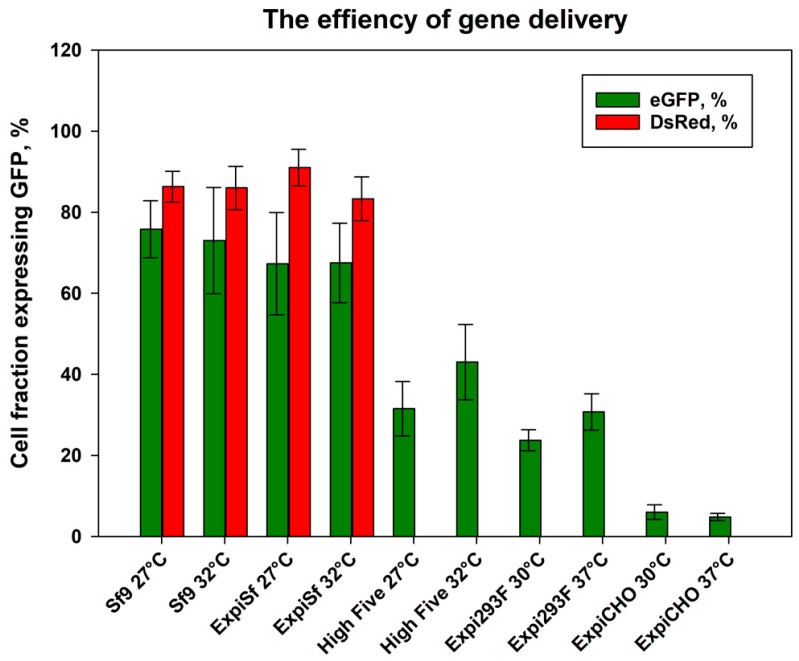
The efficiency of gene delivery. The fraction of the cells expressing GFP after viral infection/plasmid transfection is presented. The GFP fluorescence of the cells grown at different temperatures was used to estimate the fraction of the viable cells expressing GFP-fused membrane proteins to determine the efficiency of the gene delivery. The expression of the DsRed gene integrated into the genome of baculovirus was used to monitor the baculovirus infection efficiency during the virus infection phase by cytofluorometry.

**Figure 2 biomolecules-13-00817-f002:**
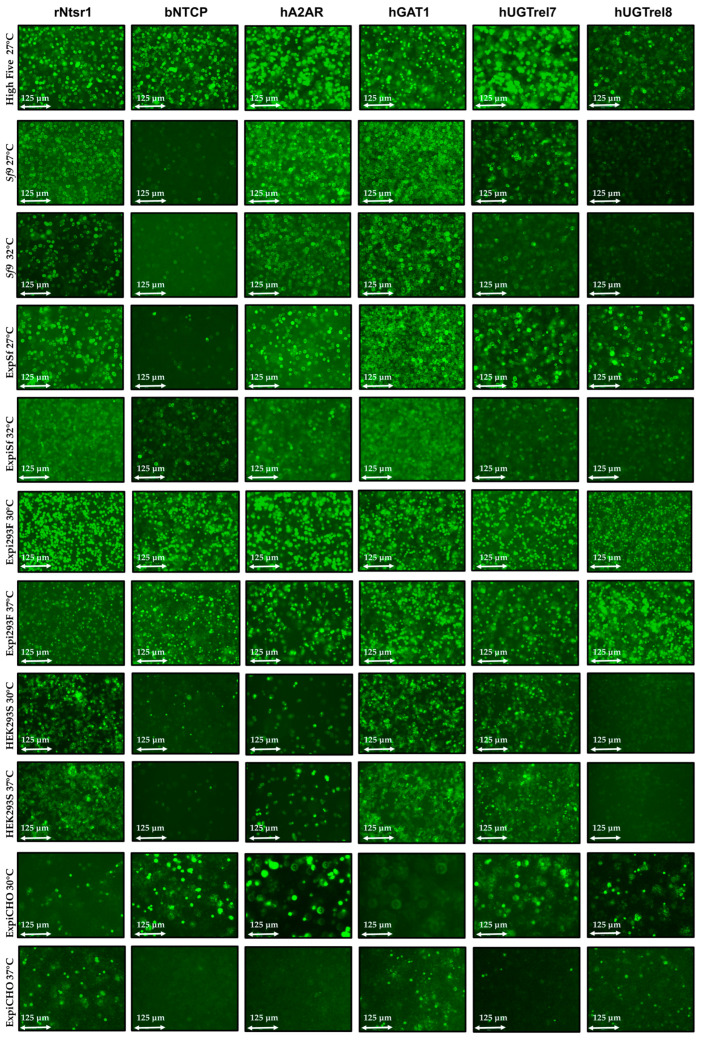
“Fluorescent micrographs” of the expression of the membrane proteins fused to eGFP in different host cell lines. Images were captured using the 20× objective of the EVOS microscope. The cells were imaged at their final densities at the final time point of the expression. While GPCR family members could be produced in most of the expression systems, the SLC transporters displayed problems to be expressed when cytotoxicity (bNTCP) or challenging sub-cellular localization (hUGTrel8) are implicated.

**Figure 3 biomolecules-13-00817-f003:**
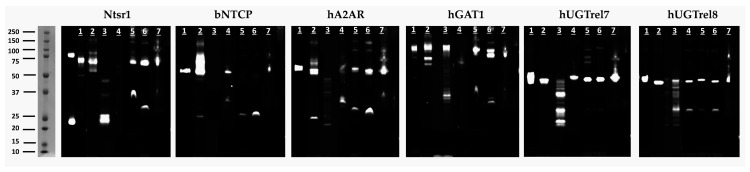
In-gel fluorescence analysis of fluorescent membrane fusion protein. The in-gel GFP signals of membrane fusion proteins expressed in different protein production systems at different temperatures after the separation of cell extracts by 1D-SDS-PAGE. The fluorescent image of the protein gels is presented. The samples of the respective expressed protein are shown in lane 1 for Expi293F (30 °C), lane 2 for Expi293F (37 °C), lane 3 for HEK293S GnTI-TetR/lentiviruses (37 °C), lane 4 for ExpiCHO (30 °C), lane 5 for ExpiSf/baculoviruses (27 °C), lane 6 for *Sf*9/baculoviruses (27 °C), and lane 7 for High Five cells (27 °C). Signals at 25 kD are due to the truncated versions of fluorescent GFP.

**Figure 4 biomolecules-13-00817-f004:**
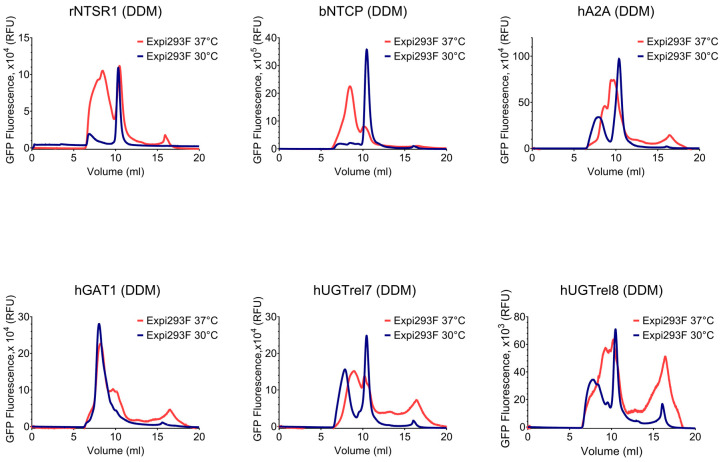
FSEC profiles of the membrane fusion proteins produced in Expi293F cells purified in DDM. Effect of the temperature on the monodispersity state of the targets. The GFP signals were normalized (Y axis) to the same scale to compare the overall retention profiles of the samples purified in DDM to correct for using membrane protein samples derived from different amounts of cells. rNTSR1 and bNTCP targets are homogeneous when purified in DDM from Expi293F cells grown at 30 °C. Other targets are heterogeneous in DDM when expressed in Expi293F cells. Expression at 37 °C promoted minor fragmentation for hA2AR and hGAT1, and severe for hUGTrel7 and hUGTrel8.

**Figure 5 biomolecules-13-00817-f005:**
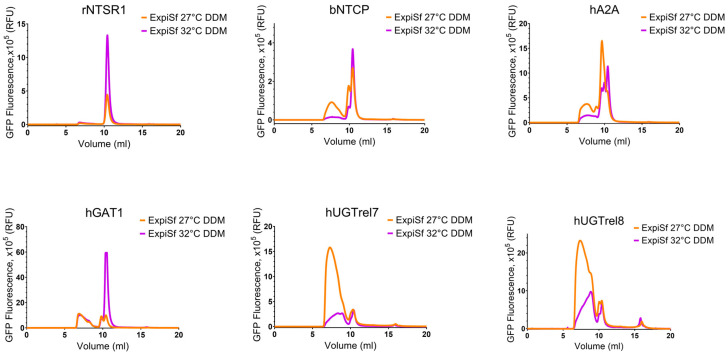
FSEC profiles of the membrane fusion proteins produced in BEVS and purified in DDM. Effect of the temperature in ExpiSf cells on the oligomerisation state of the targets. The purified samples of rNTSR1, hUGTrel7, and hUGTrel8 expressed in ExpiSf cells extracted with DDM showed, at both temperatures, monodisperse monomers for rNTSR1 on the FSEC column. bNTCP, hA2AR, and hGAT1 purified from membranes with DDM showed monodisperse peaks at 32 °C compared to weaker or polydisperse signals at 27 °C. hUGTrel7 and hUGtrel8 were only detected as polydisperse higher oligomers at 27 °C.

**Figure 6 biomolecules-13-00817-f006:**
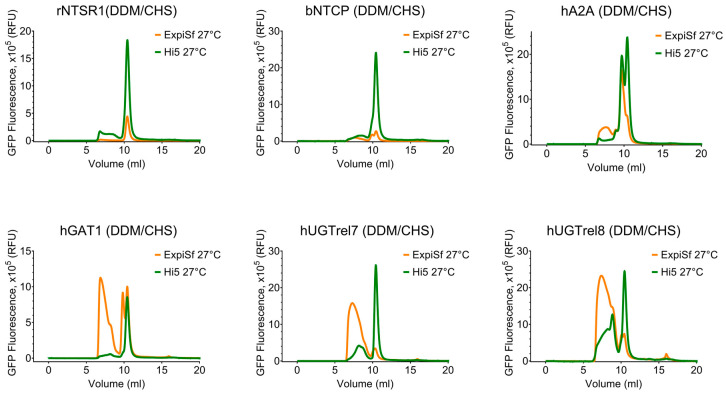
FSEC profiles of membrane fusion proteins produced plasmid-dependent TGE in High five cells at 27 °C and baculovirus-infected ExpiSf cells at 27 °C purified in DDM/CHS. Effect of the transient gene expression (TGE) on the monodispersity state of the targets. The GFP signals were normalized (Y axis) to the same scale to compare the overall retention profiles of the samples purified in DDM/CHS to correct for using membrane protein samples derived from different amounts of cells. The data for samples from ExpiSf (BEVS, 27 °C) are similar to the data in Figure 5.

**Figure 7 biomolecules-13-00817-f007:**
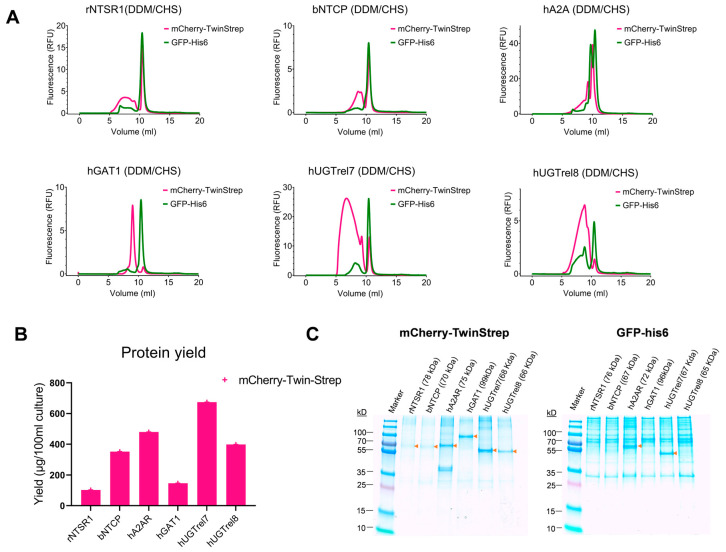
Comparison of the quality of the membrane proteins using two different fusion protein constructs expressed in High Five cells at 27 °C. (**A**): FSEC analysis of the homogeneity of the membrane fusion proteins (in DDM/CHS) expressed by TGE in High Five and purification with the indicated resin. The mCherry signals were normalized to the same scale (Y axis) as the GFP signals to compare the overall retention profiles of the membrane protein samples derived from different amounts of cells. mCherry-Twin-Strep fusion proteins purified by Strep-Tactin affinity chromatography are shown in red and eGFP-His6 fusion proteins purified by Ni-NTA affinity chromatography in green. (**B**): Protein yield in µg of protein per 100 mL culture volume obtained after Strep-Tactin purification. The purified protein for most Twin-Strep fusion proteins were higher in purity compared to His6 fusion proteins (Appendix A). (**C**): SDS-PAGE of purified membrane fusion proteins using either Twin-Strep or Ni-NTA affinity resins. The proteins were visualized using Coomassie blue. In the case of Twin-Strep purification, the proteins were prominently visible at their expected molecular size. In contrast, purification using Ni-NTA, only for the hA2AR and hUGTrel7 membrane fusion proteins, was distinctly visible.

**Table 1 biomolecules-13-00817-t001:** List of the selected target membrane proteins. The table shows the gene name, name of the protein, source of gene, UniProt ID number, and the localization of the membrane proteins: rat Neurotensin Receptor Type 1 (*Ntrs1*/rNTSR1); bovine solute carrier family 10, a sodium/bile acid cotransporter (*SLC10A1*/bNTCP); human adenosine receptor A2a (*ADORA2A*/hA2AR); human sodium- and chloride-dependent GABA transporter 1(*SCL6A1*/hGAT1); human UDP-glucuronic acid/UDP-N-acetylgalactosamine transporter (*SLC35D1*/hUGTrel7); and human UDP-N-acetylglucosamine/UDP-glucose/GDP-mannose transporter (*SLC35D2*/hUGTrel8).

No.	Gene Name	Protein Name	Source	UniProt ID	Localization
1	*Ntsr1*	rNTSR1	*Rattus norvegicus*	P20789	Plasma membrane
2	*SLC10A1*	bNTCP	*Bos taurus*	Q2KJ85	Plasma membrane
3	*ADORA2A*	hA2AR	*Homo sapiens*	P29274	Plasma membrane
4	*SLC6A1*	hGAT1	*Homo sapiens*	P30531	Plasma membrane
5	*SLC35D1*	hUGTrel7	*Homo sapiens*	Q9NTN3	ER membrane
6	*SLC35D2*	hUGTrel8	*Homo sapiens*	Q76EJ3	Golgi apparatus membrane

**Table 2 biomolecules-13-00817-t002:** Summary of the used protein production systems. The cell lines of the expression systems, the used expression media, the respective expression temperatures, and the gene delivery method are listed.

Expression System	Medium	T in °C	Expression in Days	Gene Delivery
*Sf*9	Sf-900 II SFM	27/32	3	Baculoviruses
ExpiSf	ExpiSf CD	27/32	3	Baculoviruses
High Five	EX-CELL 405	27/32	3	PEI MAX 40K
Expi293F	Expi293 expression medium	30/37	3–6	PEI MAX 40K
ExpiCHO	ExpiCHO expression medium	30/37	3–6	ExpiFectamine CHO
HEK293S GnTI-TetR	DMEM + 10% FBS	30/37	3–6	Lentivirus transduction

**Table 3 biomolecules-13-00817-t003:** The expected molecular mass of full-length membrane fusion protein (GOI-3C-eGFP-His6 construct).

No	Target Protein	Mass, kDa
1	rNTSR1-eGFP	76.3
2	bNTCP-eGFP	66.8
3	hA2AR-eGFP	73.3
4	hGAT1-eGFP	95.8
5	hUGTrel7-eGFP	67.9
6	hUGTrel8-eGFP	65.3

**Table 4 biomolecules-13-00817-t004:** Protein yield after His6-single-step affinity chromatography. Protein yields were compared by µg protein extracted from 100 mL of expression culture (see Appendix A). Protein concentrations were determined from A280 absorbance measurements using a NanoDrop microvolume spectrophotometer (Thermo Fisher Scientific, Bremen, Germany). The corresponding extinction coefficients and molecular masses were calculated using the protein sequences. However, as only a one-step purification was performed, the obtained data do not account for the potential impurities of the samples and different folding states of the obtained proteins (Appendix A). For Ni-NTA purification, all membrane fusion proteins were purified along with the host cellular impurities.

Protein Name	Estimated Protein Yield (µg Protein/100 mL Culture)					
	*Sf*9		ExpiSf		High Five	Expi293F
	27 °C	32 °C	27 °C	32 °C	27 °C	30 °C
rNTSR1	220	130	310	250	350	600
bNTCP	460	90	710	250	240	910
hA2AR	340	100	390	230	380	360
hGAT1	210	60	330	1100	90	200
hUGTrel7	610	260	940	1500	220	180
hUGTrel8	490	200	680	1700	690	50

## Data Availability

Not applicable.

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
