# Peer review of "Screening of Membrane Protein Production by Comparison of Transient Expression in Insect and Mammalian Cells"

_biomolecules, 2023, doi:10.3390/biom13050817_

Round 1
Reviewer 1 Report
The paper presents a study comparing small-scale production of six eukaryotic integral membrane proteins using insect and mammalian cell expression systems with different techniques for gene delivery. The use of GFP as a marker protein allows for sensitive monitoring of the expression. The results show that the choice of expression system has a significant impact on the yield and quality of the produced proteins. The use of virus-free transient gene expression in insect High Five cells combined with solubilization in dodecylmaltoside plus cholesterol generated the most homogeneous samples for all six targets. Additionally, the use of TwinStrep® tag for affinity purification improved the protein quality in terms of yield and homogeneity compared to His-tag purification. The study suggests that TGE in High Five insect cells can be a valuable alternative to traditional methods for producing integral membrane proteins, such as baculovirus construction and infection of insect cells or expensive transient expression in mammalian cells. The Overall work is very well execuated as per the hypothesis .
This can be accepted with minor enhancement of the pictures i.e., improve the quality of the pictures and this can be accepted in the present format.
Author Response
Dear reviewer, we wrote all the answers to the review report into the word file

Reviewer 2 Report
Kaipa et al. report on the development of a virus-free method for membrane protein production in High Five cells. The authors compare their platform to baculo infection in Sf9 cells and transfection in mammalian cells. It is not clear why they do not test baculo infection in High Five cells, which would make more sense since they use this cell line for transfection and indeed it is the most productive insect cell line. Additionally, there are several concerns on comparing their system to Expi293 and ExpiCHO cells, which are the reference platforms to produce membrane proteins nowadays. They do not do a fair comparison to none of these systems since they use unoptimized transfection methods not recommended by the manufacturer. It would be misleading for the audience to accepts this article as is now. There are several points (see below) that have to be reinforced and additional experimentation should be conducted.
Line 50) This is true, but authors should mention here that there are commercial mammalian cell lines Expi293 Gnti- that produce proteins with homogenous glycan species. This cell line is highly used to this end.
Line 54-55) This statement was true 5 years ago but it is not anymore. Mammalian cells are the most used platform to produce membrane cells with the highest yields.
Line 66-67) Authors should acknowledge studies from other groups that have worked to optimize virus-free DNA plasmid-mediated expression in insect cells.
Line 69-70) It would be interesting for the audience to discuss why authors focused on High Five cells instead of Sf21/Sf9 cells. High Five cells are generally more productive, but they have several difficulties to process some complex proteins.
Line 151) The criteria used to harvest at this timepoint should be explained.
Line 157-159) This is concerning. If authors want to compare their system with the current Expi293F system, they should use the complete Expi293 kit and not part of it. The protocol from the manufacturer clearly indicates to use Expifectamine 293 and not other transfection reagents since protein production levels can be greatly compromised. Authors should justify this and test the complete Expi293 kit for a fair comparison and not refer to reference 23, which uses a protocol that is not the one recommended by cell line manufacturers. Indeed, Expi293 manufacturers have a specific protocol for membrane protein expression and there is no reference to lowering temperature to 30 degrees. Authors should remember that they are comparing their system to the Expi293F platform, which is widely used to produce membrane proteins.
Line 216) How were ExpiCHO and HEK293S cultured and transfected? Which the DNA plasmid used for transfection of mammalian cells? This information is not provided in materials and methods.
Table 2) I do not understand why authors use Expifectamine for ExpiCHO (do authors use expifectamine CHO as recommended by the manufacturer for a fair comparison?) and do not use Expifectamine 293 for Expi293 cells?
Table 2) Why do authors use 30 degrees to culture ExpiCHO cells? It is indicated in manufacturer’s instructions to use 37 degrees always for Expi293 cells, while ExpiCHO cells can be placed at 32 degrees but not lower. These changes play a crucial role in protein expression and markedly put in a favorable position the insect cell system over mammalian cells.
Figure 1) Here is a validation of some of the points previously raised. The Expi293 and ExpiCHO cell platforms achieve a transfection efficiency of 80-90% using Expifectamine (recommended in the kit). Authors report here a transfection of 20% for Expi293 and below 5% for ExpiCHO cells with the very simple protein GFP. The way they are using both systems is very inefficient, which invalidates any comparison with the insect/cell platform.
Line 255-256) Transfection yields are quite low for High Five cells. There are several studies that report transfection efficiencies over 60%. How do authors justify this and how do they think this is something to be transferred to larger volumes?
Line 266-267) Where can the transfection levels for Expi293 at 37 degrees with GFP be found in Figure 1? I guess authors should do the same testing as they did for the other combinations to have an approximation of transfection efficiency.
Line 269-271) This is concerning. Authors should have used and followed the complete ExpiCHO kit. It is misleading for the audience to mention ExpiCHO cells are not a good system with the conditions they used…
Figure 3) Lane 7 in almost each image is cut vertically on the right. Authors should include full images. High Five cells (lane 7) usually have a larger smear above the expected protein MW in general in comparison to other conditions. How do authors explain this?
Line 301-303) On which basis do authors select 3-6 days for harvesting?
Line 314-316) How do authors explain they see higher protein degradation in ExpiSf cells at 27 degree over 32 degree? If authors check figure 5, there is generally no big difference between conditions, just probably the amount of protein produced… Which are the protein yields in both cases? Authors should provide this for all the systems they report besides transfection since in the end they are describing different production platforms.
Line 352-356) This statement is questionable since authors have not proven that High Five cells offer a better alternative to Expi293 and ExpiCHO cells in the context of using the reference TGE protocol for the latter platforms. They are comparing their optimized TGE system in High Five cells with an unoptimized system in Expi293 and ExpiCHO cells.
Line 368-371) Why do authors use different fluorescence reporters for different ligands? They should use either GFP or mCherry for a fair comparison…
Figure 7C) There is a high level of protein degradation, misfolding, or aggregation (multiple bands). How do authors know that TwinStrep (same bands can be seen but more faint) offers a better purification strategy than 6xHis? It is likely the TwinStrep tag is bound less than the 6xHis tag by the resin. Again, the same bands can be seen in general but more faint. We usually purify protein 6xHistagged with the baculo system, some of them complex proteins, and we see only one band. This confirms the protein integrity is fine and can be we shipped out. The presence of additional bands indicates that authors should substantially optimize their High Five expression platform.
Line 405-406) Do authors believe 45% is a good transfection efficiency? This implies that 55% of the cells will be consuming resources of the medium and not producing protein… Authors are strongly encouraged to increase the transfection yield to 60-70% at least. Again, Expi293 and ExpiCHO cells, when properly used, offer transfection efficiencies of 80-90%.
Line 407) In the unoptimized conditions authors used…
Author Response
Dear reviewer, we really appreciate the comments and suggestions to our manuscript, which helped us improving the manuscript substantially.

Reviewer 3 Report
In this manuscript Kaipa et. al present their findings about the comparison of membrane-associated protein expression in various cell types suitable for heterologous production of recombinant proteins for subsequent purification and analysis. They compare insect and mammalian cells transiently transfected with expression constructs using different transfection vectors (DNA, lentivirus or baculovirus) and protocols. I think this is a very important (rather) methodological paper that would help researchers interested in membrane protein synthesis and purification from eukaryotic cells. Moreover, using transient transfection makes everything easier and quicker, which is also important. However, I have serious concerns with the manuscript: the text (in particular the Material and methods section) and some experiments must be significantly revised and improved. My major concern is that the Materials and methods section is very “sketchy” and superficial, one may think that it was written quickly by several people, but not unified before submission. However, this is the main essence of the paper and other labs should rely on this section if they want to make membrane proteins. Therefore M&M must contain (literally) all the experimental details, must be consistent with the text, and must refer to catalog numbers of used reagent. I believe that after a thorough revision the manuscript will be a nice piece of scientific communication that will gain the attention of protein scientists.
My major and minor points:
Keyword: There are no “HEK” cells, but HEK293 cells (and its derivatives such as S or F, etc). HEK refers to human embryonic kidney cells. The reason why it has a number is that Frank Graham’s 293rd attempt made these cells transformed for the first time. Please do not use lab jargon in scientific publication. In the whole text HEK, ExpiHEK293, Expi293F, etc… is used. This must be revised throughout the text to HEK293 or Expi293F (Thermo/Gibco brand). And this is true for all cell lines, the nomenclature is inconsistent and must be unified.
All abbreviations must be defined for the first time. Line 45: CHO is Chinese Hamster Ovary cells (CHO). Again, HEK must be revised.
All species names when written in Latin must be italicized: e.g. line 59. Trichoplusia ni, etc….
Line 80-81: Why these membrane proteins were selected for the text. I think this should be explained in 2-3 sentences and references.
Line 87. What is DDM and CHS? Please use full names and what they are used for (ref).
The Materials and methods must be carefully revised and all experiments must be properly detailed. One may have a feeling that this section was written by several people, but was not unified.
Provide catalog numbers and vendors (in parenthesis) of all reagents (including cell lines). “Obtained from Invitrogen” is not enough. Correctly: “was obtained from Abcam (cat# 1234, Cambridge, UK)”.
Provide catalog numbers for labware and even equipment (e.g. orbital shaker, sonicator, etc.). For example sonication is nicely detailed, but we do not know which machine or sonication tip should be used at “power 10 %”. However, this is an extremely important step to extract proteins from membranes.
Concentrations must be revised. e.g. line 106: I guess the authors used 5x ten to the sixth (hence 5 million) cells in 1 ml culture, and not 5x106, which is 530 cells/ml. Please revise everywhere.
Avoid using lab jargon and define all reagents properly. Thermo FS does have Express media, but this is not for Expi293F cells. For these cell we use “Expi293 Expression medium”. Try to be 100 % accurate.
Follow the nomenclature standards of chemical compounds. e.g. CO2 and MgCl2: “2” must be written in subscript.
Experimental temperatures are not defined! This is also very important for reproducibility. e.g. line 143. What is the centrifugation temperature? Please revise throughout the text (centrifugation, incubation, purifications, etc.).
Line 145: again we don’ know how an important reagent was made. PEI: what is PEI, how was the stock made and in what solvent and what concentration, etc. Please revise.
Line 162-163: How and why this buffer composition boost expression. Please explain and cite previous publications appropriately.
Line 172: what type of ultrasound machine and tip was used?
ExpiCHO is not mentioned in the M&M. Again it should be unified: ExpiCho or Expi-CHO or CHO.
Table 2: It shows that Expi293F and ExpiCHO cells were grown at 30 C. However, in the M&M section the growing temperature is 37 C. Even Thermo suggests 37 C for cultivation of these cell types and we and other labs also use 37 C. Why is the discrepancy between the M&M and Results, and why the authors decided to use non physiological temperature for mammalian cells? Moreover, these cells prefer higher CO2 concentration, therefore I believe that the lower expression of the test proteins in these cells is due to the non-optimal conditions. These experiments must be repeated at 37 C in all cases (major concern).
Lentiviral transfection is not detailed in the M&M section.
Figure 1.: figure captions are different than in the text? Please unify. Again, there are no Expi-HEK-293F cells.
Line 251: this paragraph says that the expression was lower in mammalian cells. I am sure this is due to the low temperature (30 C). Please repeat at 37 C!
Line 252: what does “multiple experiments” mean? How many exactly? n=??
Line 256: “30-45%”: where it is shown? Refer to the figure please.
Figure 2: Unify figure captions (ExpiHEK…, etc). This is clearly not a “Heatmap”. These are microscopic images. Please revise appropriately. The figure says that ExpiCHO cell were grown at 32 C. Previously in table 2 it was 30 C. Please clarify this and detail the ExpiCHO experiments in the M&M section.
Figure 2 and Suppl F1: Scale bars are not visible at all. Please revise in the figures and also indicate the size of scale bars in the figure legends.
Line 269: Perhaps the “expression level” of GFP-fusions detected by fluorescent microscopy is misleading. It may degrade (through the ERAD) or not folded properly. To prove the above statement, authors should do a western blot using whole cell lysates and anti-GFP antibody, and/or support their statement by QPCR. Either WB or QPCR should be done during the revision to support the above.
Figure 7C: On the stained gels it is not obvious (in all cases) which is the purified proteins, because of the high background of non-specifically purified proteins. The indicated (with arrowhead) weak bands should be validated by western blot or mass spectrometry.
Please revise all figure/table legends. The title is not enough with short description. We should be able to understand the figure by reading the legends only.
Author Response
Dear reviewer, we are very gratefull for all the suggestions and comments to our manuscript. We used the questions to improve the manuscript

Reviewer 4 Report
In this manuscript, the production of six membrane proteins has been evaluated in six eukaryotic cell lines, including insect- and mammalian cells. Production of recombinant membrane proteins has been notoriously difficult and identifying the bottlenecks of the production is essential to isolate good quality proteins for structural and functional characterization. The purpose of this study is to examine the quantity and quality of recombinant membrane proteins by screening small-scale production experiments. Various parameters have been analyzed, including transfection method, cell line, temperature and purification tag.
Overall, the manuscript is well written and contains detailed description of the experimental procedures. The Authors have performed an extensive characterization of the recombinant membrane proteins in the various expression systems using fluorescence-based methods. The results are clear and presented in nice figures. I do support the conclusion that the TGE method for production of membrane proteins fused to a fluorescent reporter and a Twin-StrepII tag in High Five cells is an excellent system to analyze production of recombinant membrane proteins.
I only have two minor comments for the Authors to address:
1) Line 67: ‘.. until the TGE method was largely improved by Bleckmann et al [14} using High-Five cells.’ This reference [14] does not describe TGE in High-Five cells. I suppose the Authors want to refer to the other publication by Bleckmann et al., nr [22] where the TGE method in High Five cells is described.
2) Expression of the membrane proteins was performed at different temperatures; expression in Expi293F cells was performed at a ‘default’ culturing temperature of 37 °C and at reduced temperature (30 °C) to examine mono/poly-dispersity of the protein. I can see the reason to reduce the temperature to improve folding of the protein. However, for the Expisf cells both the ‘default’ temperature and an increased temperature (32°C) was used. Although the higher temperature promoted purification in DDM, could the Authors explain why a higher temperature was chosen rather than a reduction in temperature as this is not mentioned in the manuscript.
Author Response
Dear reviewer, we thank you for the kind answer and points you raised. The reply is in the reply letter to the reviewer

Round 2
Reviewer 2 Report
Authors have failed to address the major concerns of this study. There are different points that still need to be addressed.
Authors comment) We thank the reviewer for the raising this option. The used systems were chosen on the basis of the state of the art at the time the project started. We already used many frequently used successful expression systems. We consider the High Five BEVS system for further scale up for purifying large amounts of membrane protein, but not for the initial screening
Authors should include an experiment of infection of High Five cells with the baculovirus system since it is the standard platform to produce recombinant proteins in insect cells and also matches the system they propose, protein production in High Five cells by plasmid DNA transfection. This way, there would be a head to head comparison of High Five cells with transfection and baculovirus infection. In any case authors prove that transfection of insect cells is better in terms of protein production than baculovirus infection since different cell lines are used (i.e ExpiSf vs High Five) and comparisons are misleading.
Authors comment) According to the Protein Data Bank (2022) the two expression systems most often used for producing purified membrane proteins for structural studies are baculovirus infected insect cells (21.6 %) and HEK293 mammalian cells (25.5 %), hence our focus on these systems. Although CHO cells are widely used for functional studies, ExpiCHO do not appear to be widely used for the production of recombinant membrane protein expression. We note that in the information provided by the commercial supplier, Expi293 cells are recommend for expression screening while ExpiCHO cells are more appropriate for scale-up and manufacturing (https://assets.thermofisher.com/TFS-Assets/BID/posters/high-titer-protein-expression-expiCHO-expi293-poster.pdf). Our aim was not to compare directly two commercially available systems but rather to use optimized procedures for transient gene expression in HEK293 cells as most laboratories that involves changing to PEI 40 dependent transfection to reduce the costs for using the commercial system. This is common practice in most expression facilities. Transfection efficiencies of 50% are sufficient for comparing the systems. However, for the ExpiCHO system we used the protocol and reagents from the commercial supplier. We have now set this out in the methods and materials section (lines 208-222).
Authors still defend that using the Expi293 and ExpiCHO systems with unoptimized transfection conditions (again, manufacturers DO NOT recommend this), which is not acceptable. There are two options to move forward, either authors repeat the experiments using the conditions recommended by manufacturers or authors withdraw the mammalian cell comparison part. It is misleading for the audience to do an “unfair” comparison of these production platforms, considering that Expi293 cells are the most widely used platform for membrane protein expression. Authors argue the cost of the bioprocess, but using optimized transfection conditions improves the yield and the cost of protein/volume of medium used decreases. Authors should keep this in mind. Seeing the comments of the other reviewers, it is also argued that additional experiments should be done using the optimal conditions of mammalian cell systems authors test.
Authors comment) We added additional references of the most important groups involved in establishing the robust TGE in High Five insect cells
There are significant studies missing (Europe, Japan) of other research groups pioneering transfection of High Five cells.
Authors comment) Successful transient gene expression (TGE) using pOpIE2Plasmids could only be performed efficiently in High Five insect cells till date. Attempts were made by us and other researchers to optimize the TGE system in Sf9/SF21 cells, however the efficiency and yield was very poor.
This is not true. There are several studies addressing transfection of Sf9 cells in suspension conditions that yield higher transfection efficiencies than what the authors report in this study for High Five cells. Authors need to still justify why they use High Five cells over Sf9 cells, especially since one of the comparators they use is Sf9 cells and the baculovirus system.
Authors comment) We thank the reviewer for showing his concerns but PEI assisted transfection of HEK293 cells has been optimised and is generally well working in many laboratories. The compromise to use PEI instead of Expifectamine 293 is based on economic reasons. Therefore, our aim is to compare optimized protocols for PEI-mediated transfections of Expi293F and High Five cells, that will give people an opportunity to try other protocols for protein production in case their target is challenging and requires extremely expensive protocols according to manufacturer recommended settings.
Other reviewers and I insist on repeating the mammalian cell transfection experiments using the optimized conditions. Again, PEI is NOT recommended to use with Expi293 cells, which results in poor transfection/production yields. Either authors redo the experiments, or they should withdraw the mammalian cell experiments part.
Authors still fail to include the DNA plasmid vector used for protein expression in mammalian cells in the materials and methods section.
Authors still fail to include a readable scale bar for each Figure in Figure 2. Clearly, the zoom in Is higher for High Five cells in comparison to other systems. It seems the authors want to show less total cells for High Five cells. Authors should use the same scale bar and that the scale bar is readable.
Authors comment) We have data on protein production yields and included these data in Supplementary Materials Supplementary Table T1.
Authors should put this information in the manuscript as it is highly relevant, not in supplementary files.
Authors comment) We aimed to show that PEI assisted transfection in High Five cells is at least comparable to PEI transfection of HEK293F cells. Both protocols will find their users since cheaper and faster (High Five cells) expression screening of integral membrane proteins constructs is limiting factor. Additionally, we did not compare the ExpiCHO expression system further for our FSEC analysis. For the initial screening we did perform the transfection experiment as described by the supplier.
Yes, but it seems that authors fail to understand that Expi293 cells are a subclone of HEK293 cells that has been optimized to be transfected with Expifectamine293, NOT with PEI. So this is an unfair comparison that benefits the High Five system. Following the same criteria, authors should test transfection of High Five cells with Expifectamine293…
Authors comment) We did use a comparison of both fluorescent tags eGFP-His and mCherry-Twin-Strep for all the target proteins.
It does not seem this is the case since in figure 1 there are no transfection results for all conditions with DsRed.
Authors comment) As it was a one-step affinity purification, it is obvious that we cannot expect very pure protein samples. However, purification using Twin-Strep-tag purification yields relatively more pure samples already with first round of purification compared to His6-tag purification. This was the reason why we used Fluorescence bases Size exclusion chromatography (FSEC) to analyse the homogeneity of the purified membrane protein samples but not absorbance at 280nm. Additionally, the back ground in the denaturing SDS-PAGE gel is due to the contamination with host The SDS-PAGE gels of individual purification performed using TwinStrep (supplementary Figures S4 and S5) exhibits clearly less host back ground bands compared to His6 purification. This is not due to degradation of the protein.
This is even worse, affinityTag purification should remove almost all host cell proteins. Again, we use HisTag purification for our proteins and only see ONE band that correspond to the protein of interest after validation with western blot. Which are the conditions authors use that result in so much host cell protein co-purification? His and Streptag purification strategies are highly specific. Authors should consider revising their purification protocol. It is not scientific to mention that there is only one purification step (as implying that more steps should be included) when most of the people in the field purify proteins by affinity chromatography (mostly 6xHis tag) and only use ONE purification step. The affinity step is enough for the protein to be used in serology, animal experiments, and crystallography assays (please check studies from the Scripps on how they purify proteins for crystallography studies, which have to be really pure).
Authors comment) We thank again the referee for all the important comments. The protocol we use for counting number of cells expressing GFP (targets fused to GFP) eliminates from the final evaluation measurements fractions of fragmented, non viable cells and eliminates autofluorescent cells. This gives generally lower transfection efficiencies compared to other published data which use all fluorescent counts. Therefore, it provides accurate estimation of transfection efficiencies for the plasmids with membrane protein targets. Since we compare all systems on one scale, relative difference is more speaking than the absolute values. We consider efficiencies as more than 50% as sufficient and normally reach >80% transfection with the eGFP control which is an easy to express protein. Reaching 45 % transfection efficiency for membrane proteins expression is sufficient to do purification and scale-up for the required amount of protein in the cheap and robust High-Five insect cell plasmid-based transient gene expression.
This is not completely true. Authors obtain similar transfection yields of Expi293 cells and High Five cells. If authors used the optimized transfection system for Expi293 cells, they would probably double or triple the transfection efficiency of Expi293 cells. Again, 45% transfection is a low transfection percentage for any protein. Still, I do not see the advantage of using High Five cells over ExpiSf cells at 27 degree. The yields are mostly higher in ExpiSf cells (supplementary figure 1), but these results do not agree with the RFU shown in figure 6.
Author Response
Dear Reviewer 2,
we have replied to your suggestions and comments in the "author-cover lettrer 27239967.1"

Reviewer 3 Report
Na
Author Response
Thank you for accepting the paper
Round 3
Reviewer 2 Report
Authors still fail to address the main concerns of this study and they refuse to provide additional information. There are four major points that still need to be addressed:
- - Authors acknowledge that transfecting Expi293 cells with PEI 40kDa is not the best strategy to attain a high level of transfection, since transfection of these cells has been optimized with Expifectamine293. This should be indicated in the results and discussion sections since the optimized system would have resulted in higher expression yields. The same should be mentioned for the ExpiCHO system.
- - I am still confused about why the authors do not test the expression of these proteins in Trichoplusia ni cells with baculovirus. They argue that Sf9 cells are more frequently used to produce membrane proteins but the method they present is based on transfection of Trichoplusia ni cells. They should include a fair comparator with the Trichoplusia ni baculovirus system, at least for some of the proteins to show how their system compares with the workhorse platform for recombinant protein production.
- - Relevant studies on the use of transfection of Sf9 cells in suspension conditions for recombinant protein expression are still missing.
- - Authors should include control experiments with simpler proteins to show the transfection yield/production of the systems they test. They mention they usually get >60% for intracellular soluble eGFP. This would reinforce the point that they see a remarkable decrease in transfection yield/production when they try to express complex proteins, but that the protocols they test are optimized.
Author Response
Dear Editors and Reviewers,
We gratefully accept your offer to revise the manuscript we submitted to Biomolecules entitled “Screening of membrane protein production by comparison of transient expression in insect and mammalian cells”
Point 1 from reviewer 2
"Authors acknowledge that transfecting Expi293 cells with PEI 40kDa is not the best strategy to attain a high level of transfection, since transfection of these cells has been optimized with Expifectamine293. This should be indicated in the results and discussion sections since the optimized system would have resulted in higher expression yields. The same should be mentioned for the ExpiCHO system."
My comment:
I think that the authors could easily handle this comment, since the reviewer only ask to be careful with the conclusions.
Answer: I have modified the text in the introduction, materials and methods, results, discussion and the final conclusion to address the request of the reviewer 2 to take into account that the transfection method using Expifectamine 293 may result in higher transfection efficiencies. In my new version these changes are in line 29-30; 92-103; 206-209; 378-381; 565-569; 638-641. The ExpiCHO was used with all the materials as described by the manufacturer as we describe in the materials and methods.
Point 2 from reviewer 2
"I am still confused about why the authors do not test the expression of these proteins in Trichoplusia ni cells with baculovirus. They argue that Sf9 cells are more frequently used to produce membrane proteins but the method they present is based on transfection of Trichoplusia ni cells. They should include a fair comparator with the Trichoplusia ni baculovirus system, at least for some of the proteins to show how their system compares with the workhorse platform for recombinant protein production."
Academic editor: …..An interesting parallel to bacteria, though, can be seen here: for membrane proteins, investigators prefer to use the BL21DE3 (non star) even though it gives lower yields than the "star" version, because membrane proteins are generally toxic, and too much production may cause lower yields. Is there a similar problem for membrane proteins in insect cells transformed with baculovirus v/s transfection? This could explain the choice to not test the most efficient system.
Answer to the suggestion of the academic editor: This is a good point as over-expression of membrane proteins in insect cells using baculoviruses may also lead to toxic effects irrespective of the cell line. As our study focused on screening at small-scale, we decided to use the Sf cell line to ensure a streamlined workflow. Here you can transfect and do additional expression testing in only one (maximally two steps). To use High Five, you first have to generate and accumulate the Baculoviral particles in Sf cells and only thereafter you are able to test expression in High Five. This requires more steps and increases the timeline for expression screening. Once BEVS is chosen as the optimal production system, you should try definitely whether infection of High Five could increase you yield and look for the effect on the quality of the membrane protein, but this is not the aim of this manuscript
Point 3 from reviewer 2
"Relevant studies on the use of transfection of Sf9 cells in suspension conditions for recombinant protein expression are still missing."
Academic editor
I am not familiar with Sf9 cells, but, since they are widely used cells, I guess that relevant literature should be rather easy to find.
Answer: We carried out a further extensive literature search from which we identified a few suitable references. [6] Geisse et al., [18] Loomis et al., [21] Shen et al. (2014) and [25] Farell et al. Within these references and our own paper [28] we have all the relevant references, dealing with development of the Virus-Free/Plasmid-dependent Transient Gene Expression in Sf and Hi5 cell lines, included into this manuscript. The positions are marked.
Point 4 from reviewer 2
"Authors should include control experiments with simpler proteins to show the transfection yield/production of the systems they test. They mention they usually get >60% for intracellular soluble eGFP. This would reinforce the point that they see a remarkable decrease in transfection yield/production when they try to express complex proteins, but that the protocols they test are optimized."
Academic editor
This is a somewhat parallel comment to Point 2, about the difference between soluble v/s membrane proteins. The reviewer asks for the production of a basic soluble protein (eGFP) to test the different cells, but this is likely not relevant to the goals of a manuscript focusing on membrane proteins. From my experience in bacteria/yeast, I know that the use of a simple eGFP protein does not bring any relevant information about the production of membrane proteins.
Answer: We fully agree with the view of the editor. We have established the High Five TGE as described in e.g. [22,24,28] For these experiments we used eGFP, and intracellular proteins. In those publications, we show which yields and transfection efficiencies were reached. In the current manuscript we show for the difficult-to-express membrane proteins the protein yield was expected to be low, but as we show it still works sufficiently well
In summary we have taken Points 1 and 3 into account and changed the manuscript to address the comments. We have also added a few lines to explain our choices for the expression systems tested in this manuscript for screening of membrane protein production systems.
We greatly appreciate your effort to improve the manuscript. We are looking forward to the comments to finalize the submission for this manuscript.
Sincerely yours,
Dr. Joop van den Heuvel
Helmholtz Centre for Infection Research
